# Genetic evaluation of growth rate and efficiency-related traits in Menz sheep

Asfaw Bisrat[1], Shanbel Besufkad[1]*, Aschalew Abebe[1], Shenkute Goshme[1], Ayele Abebe[1], Tesfaye Zewdie[1], Alemnew Areaya[1], Zeleke Tesema[1], Chekol Demis[1], Erdachew Yitagesu[1], Yeshitila Wondifra[1], Tadiwos Asfaw[1], Enyiew Alemnew[1], Solomon Gizaw[2], Tesfaye Getachew[3], Barbara Rischkowsky[3], Mourad Rekik[4], Berhanu Belay[3], Aynalem Haile[3]

1 Amhara Agricultural Research Institute, Debre Birhan Agricultural Research Center, Debre Birhan, Ethiopia, 2 International Livestock Research Institute (ILRI), Addis Ababa, Ethiopia, 3 International Centre for Agricultural Research in the Dry Areas (ICARDA), Addis Ababa, Ethiopia, 4 International Center for Agricultural Research in the Dry Areas (ICARDA), Tunis, Tunisia

* shanbel21@gmail.com

## Abstract

Growth efficiency traits are economically important and their genetic improvement is essential in small ruminants, particularly under conditions of limited feed availability. Data on body weights of Menz sheep collected from 2009 through 2023 in Menz sheep community-based breeding programs (CBBPs) were used to estimate genetic parameters for six months weight (WT6), average daily gain from birth to weaning (ADG1), average daily gain from weaning to six months (ADG2), and average daily gain from six months to yearling (ADG3), and corresponding kleiber ratios (KR1, KR2, KR3), efficiency of growth (GE1, GE2, GE3), and relative growth rate (RGR1, RGR2, RGR3). Least-squares analysis was performed using the PROC GLM procedure in SAS 9.4. (Co)variance components and corresponding genetic parameters were estimated using the Average Information Restricted Maximum Likelihood (AI-REML) method in WOMBAT software by fitting six univariate animal models that incorporated various combination of direct additive, maternal genetic, and permanent environmental effects. The year of birth, season and villages of CBBP contributed significantly ($P < 0.001$) to the variation of all the considered traits. The model with direct additive genetic effect alone provided the best fit for all traits except WT6 and ADG1, for which the model including direct and maternal genetic effects performed best. Direct heritability estimates ($h^2_a$) were $0.46 \pm 0.02$, $0.44 \pm 0.02$, $0.35 \pm 0.02$, $0.21 \pm 0.02$, $0.38 \pm 0.02$, $0.36 \pm 0.02$, $0.23 \pm 0.02$, $0.39 \pm 0.02$, $0.39 \pm 0.02$, $0.48 \pm 0.03$, $0.36 \pm 0.02$, $0.42 \pm 0.02$, and $0.47 \pm 0.03$ for WT6, ADG1, ADG2, ADG3, KR1, KR2, KR3, GE1, GE2, GE3, RGR1, RGR2, and RGR3, respectively. Maternal heritability ($h^2_m$) for WT6 and ADG1 was $0.02 \pm 0.01$ and $0.02 \pm 0.01$. Moderate to high direct heritability estimates for the studied traits revealed that direct selection would be yield significant genetic progress. Genetic correlation estimates between the traits raged

**Data availability statement:** All relevant data are within the paper and its Supporting Information files.

**Funding:** The author(s) received no specific funding for this work.

**Competing interests:** The authors have declared that no competing interests exist.

from −0.66 for ADG1-RGR2 and KR1-RGR2 to 0.98 for KR2-RGR2. Traits measured within the same growth period showed strong positive genetic and phenotypic correlations. Since Menz sheep are raised under harsh environmental conditions, incorporating efficiency related traits into the selection program can enhance the overall efficiency of the breeding program. Selection for WT6 would be more effective and increase efficiency-related traits due to their positive genetic correlations.

## Introduction

Small ruminants play a crucial role in supporting the livelihoods of smallholder farmers and are widely distributed across Ethiopia's diverse ecological zones [1]. Menz sheep is an indigenous breeds native to the central highlands of Ethiopia. The breed is adapted to the high altitude terrain and is known for its tolerance to drought and endo-parasite infections [2,3]. The breed thrives under harsh conditions of feed scarcity, high disease prevalence, and limited crop production caused by extreme low temperatures. As a result, smallholder farmers rely on Menz sheep as a key component of their livelihoods [4,5].

Besides its economic value and adaptability to harsh environments, the productivity of Menz sheep remains low compared to other Ethiopian indigenous breeds [6–8]. Since 2009, community-based breeding programs (CBBPs) have been implemented as a promising strategy to improve Menz sheep within low-input smallholder production system [9,10]. Farmers have prioritized Improving growth rate, survival and lambing interval while maintaining the breed's environmental adaptations [8,11]. CBBPs have led to significant improvement in live body weight, body conformation, and overall animals appearance [8,12–14].

Selection for increased body weight in sheep is typically associated with higher nutritional requirements. Since Menz sheep is raised in area prone to feed scarcity, incorporating growth efficiency traits into selection programs is essential for sustaining breeding program. However, direct selection for this trait is challenging due to the lack of individual feed intake records, particularly in low input production system. Kleiber ratio provides an indirect measure of feed conversion efficiency and can be easily applied in any production system [15]. Animals with higher kleiber ratio values are considered more efficient in feed utilization [16]. Furthermore, selection based on kleiber ratio has been suggested as an efficient selection criterion for improving feed efficiency and growth traits in low input production system [17]. To achieve optimum genetic progress in a selection program, knowledge of genetic parameters regarding growth and efficiency related traits are needed. To date, no studies have estimated the genetic parameters of efficiency traits in Menz sheep. Therefore, this study aimed to estimate genetic parameters of growth rate and efficiency related traits in Menz sheep under community-based breeding programs (CBBPs).

## Materials and methods

### Location, flock management and data collection

This study utilized data from community-based breeding programs (CBBPs) of Menz sheep in the central highlands of Ethiopia, and no ethical concerns were associated

with the data collection. The Menz area lies between 39-40°E longitude and 10-11°N latitude, at altitude range of 2500–3200 meter above sea level. The area is characterized by a mixed subalpine sheep-barley production system [1]. The Menz sheep CBBPs were implemented in two district, Menz Gera and Menz Mama, located 280 and 240 km from Addis Ababa respectively. During the study period, the mean minimum and maximum temperatures were 2.21 °C (December) and 23.84 °C (January). The mean annual rainfall during the study period was 891 mm, with a bimodal distribution, of which 75% occurred from June to September, while the short rains were received in March-April. The flock primarily grazed on low-quality communal grazing lands and crop aftermath, with crop residues provided during dry season. Limited feed availability, lack of improved grazing land management and insufficient development of improved forages have contributed to poor productivity of the flock [5].

Lambing in Menz sheep occurs year-round, with peak seasons in December, September, and January while the lowest lambing are observed in March and April [14]. New born lambs were weighed and ear-tagged at birth or within 24 hours thereafter. Weaning, six-months and yearling weight were recorded at approximately 90, 180, and 365 days of age, respectively, with a tolerance of plus or minus five days, and subsequently linearly adjusted to the standard ages of 90, 180, and 365 days. Breeding rams were selected based on six-months weight estimated breeding values (EBVs) along with aesthetic traits such as horn shape, tail type, coat color, and body conformation. A single sire mating system was employed, with one selected breeding ram allocating for 25–30 ewes by organizing the farmers into ram groups according to proximity. To minimize inbreeding, rams were carefully assigned to each ram group and rotated annually across different groups [14].

## Data analysis

The dataset comprised 22,712 lamb records from 6,528 ewes sired by 586 rams, born between 2009 and 2023. Body weight records at birth (WT0), weaning (WT3), six months of age (WT6), and yearling (WT12) were used to drive the traits of interest. Growth traits analysed included six months weight (WT6), average daily gain from birth to weaning (ADG1), from weaning to six months (ADG2), and from six months to yearling (ADG3); kleiber ratio from birth to weaning (KR1), from weaning to six months (KR2), and from six months to yearling (KR3); efficiency of growth from birth to weaning (GE1), from weaning to six months (GE2), and from six months to yearling GE3) and relative growth rate from birth to weaning (RGR1), from weaning to six months (RGR2), from six months to yearling (RGR3). The summery of the data is presented in Table 1.

Average daily gain from birth to weaning (ADG1), from weaning to six months (ADG2) and six months to yearling (ADG3) was calculated as $\frac{(WT3-WT0)}{90 \times 1000}$, $\frac{(WT6-WT3)}{90 \times 1000}$, and $\frac{(WT12-WT6)}{180 \times 1000}$, respectively.

Kleiber ratio from birth to weaning (KR1), from weaning to six months (KR2) and six months to yearling (KR3) was calculated as $\frac{ADG1}{WT3^{0.75}}$, $\frac{ADG2}{WT6^{0.75}}$, and $\frac{ADG3}{WT12^{0.75}}$, respectively.

Growth efficiency quantifies the weight gain of an animal relative to its initial body weight over a given period. Efficiency of growth from birth to weaning (GE1), from weaning to six months (GE2) and from six months to yearling GE3) was calculated as $\frac{(WT3-WT0)}{WT0 \times 100}$, $\frac{(WT6-WT3)}{WT3 \times 100}$, and $\frac{(WT12-WT6)}{WT6 \times 100}$, respectively.

Relative growth rate from birth to weaning (RGR1), from weaning to six months (RGR2) and from six months to yearling (RGR3) was calculated as $\frac{[\ln(WT3)-\ln(WT0)]}{90}$, $\frac{[\ln(WT6)-\ln(WT3)]}{90}$, and $\frac{[\ln(WT12)-\ln(WT6)]}{180}$ [15,18].

Least-squares analysis was performed using the PROC GLM procedure of SAS software [19] to determine the significant environmental effects. The fixed effects included in the model were year of birth (14 levels), village of CBBP (five levels), sex (male or female), type of birth (single or twin), parity of the dam (1–9 parities), and lambing season (main rainy, dry or short rainy). Least squares means for factors with more than two levels were separated using the Tukey-Kramer test. The model for the analysis of fixed effects was as follows

$$Y_{ijklmno} = \mu + Y_i + V_j + S_k + BT_l + P_m + Sn_n + e_{ijklmno} \tag{1}$$

**Table 1. Characteristics of data structure.**

| Parameters | Traits | | | | | | | | | | | | |
|---|---|---|---|---|---|---|---|---|---|---|---|---|---|
| | WT6 | ADG1 | ADG2 | ADG3 | KR1 | KR2 | KR3 | GE1 | GE2 | GE3 | RGR1 | RGR2 | RGR3 |
| No. of animals | 19470 | 22712 | 18826 | 12012 | 22712 | 18826 | 12012 | 22712 | 18289 | 11207 | 22712 | 18289 | 11207 |
| No. of records | 15697 | 19075 | 15142 | 8981 | 19075 | 15142 | 8981 | 19075 | 14617 | 8199 | 19075 | 14617 | 8199 |
| Sire[a] | 553 | 586 | 550 | 460 | 586 | 550 | 460 | 586 | 542 | 440 | 586 | 542 | 440 |
| Sire[b] | 322 | 405 | 312 | 203 | 405 | 312 | 203 | 405 | 296 | 167 | 405 | 296 | 167 |
| NPR/Sire | 26.74 | 32.55 | 27.53 | 19.52 | 32.55 | 27.53 | 19.52 | 32.55 | 26.97 | 18.63 | 32.55 | 26.97 | 18.63 |
| Dam[a] | 5791 | 6528 | 5613 | 3649 | 6528 | 5613 | 3649 | 6528 | 5475 | 3450 | 6528 | 5475 | 3450 |
| Dam[b] | 2295 | 3080 | 2218 | 1034 | 3080 | 2218 | 1034 | 3080 | 2108 | 896 | 3080 | 2108 | 896 |
| NPR/Dam | 2.71 | 2.92 | 2.7 | 2.46 | 2.92 | 2.7 | 2.46 | 2.92 | 2.67 | 2.38 | 2.92 | 2.67 | 2.38 |
| Mean | 12.55 | 65.93 | 49.93 | 44.4 | 12.92 | 7.43 | 4.71 | 244.97 | 56.2 | 57.91 | 1.32 | 0.47 | 0.24 |
| S.D. | 2.48 | 25 | 30.12 | 26.81 | 2.54 | 4.16 | 2.64 | 110.69 | 36.53 | 32.53 | 0.34 | 0.25 | 0.11 |
| CV (%) | 15.88 | 31.48 | 55.08 | 53.4 | 15.98 | 50.97 | 51.04 | 36.86 | 57.49 | 44.49 | 20.76 | 46.07 | 36.53 |

[a] Number of sires and dams with progeny, [b] Number of sires and dams with progeny and record, NPR: Average number of progeny with records, S.D.: standard deviations, CV.: Coefficient of variation, WT6: six months weight, ADG1: average daily gain from birth to weaning, ADG2: average daily gain from weaning to six months, ADG3: average daily gain from six months to yearling, KR1: kleiber ratio from birth to weaning, KR2: kleiber ratio from weaning to six months, KR3: kleiber ratio from six months to yearling, GE1: growth efficiency from birth to weaning, GE2: growth efficiency from weaning to six months, GE3: growth efficiency from six months to yearling, RGR1: relative growth rate from birth to weaning, RGR2 relative growth rate from weaning to six months, RGR3: relative growth rate from six months to yearling

Where $Y_{ijklmno}$ is an observation for growth rate and efficiency-related traits; $\mu$ is overall mean; $Y_i$ is fixed effect of birth year; $V_j$ is fixed effect of village of CBBP; $S_k$ is fixed effect of lamb sex; $BT_l$ is fixed effect of birth litter size; $P_m$ fixed effect of dam parity; $Sn_n$ is fixed effect of birth season and $e_{ijklmno}$ is residual error.

The (co)variance components and corresponding genetic parameters for the studied traits were estimated using Average Information Restricted Maximum Likelihood (AI-REML) using WOMBAT software [20]. Six univariate animal models, incorporating different combination of direct additive, maternal genetic and permanent environmental effects, were used for the genetic analysis. Genetic and phenotypic correlations were estimated through multivariate analysis. The models applied were as follows:

$$\text{Model (1) } y = \mathbf{X}\beta + \mathbf{Z}_a\alpha + e \tag{2}$$

$$\text{Model (2) } y = \mathbf{X}\beta + \mathbf{Z}_a\alpha + \mathbf{Z}_{pe}pe + e \tag{3}$$

$$\text{Model (3) } y = \mathbf{X}\beta + \mathbf{Z}_a\alpha + \mathbf{Z}_m m + e \quad Cov(\alpha, m) = 0 \tag{4}$$

$$\text{Model (4) } y = \mathbf{X}\beta + \mathbf{Z}_a\alpha + \mathbf{Z}_m m + e \quad Cov(\alpha, m) \neq 0 \tag{5}$$

$$\text{Model (5) } y = \mathbf{X}\beta + \mathbf{Z}_a\alpha + \mathbf{Z}_m m + \mathbf{Z}_{pe}pe + e \quad Cov(\alpha, m) = 0 \tag{6}$$

$$\text{Model (6) } y = \mathbf{X}\beta + \mathbf{Z}_a\alpha + \mathbf{Z}_m m + \mathbf{Z}_{pe}pe + e \quad Cov(\alpha, m) \neq 0 \tag{7}$$

Where y is vector of observations for the studied traits (WT6, ADG1, ADG2, ADG3, KR1, KR2, KR3, GE1, GE2, GE3, RGR1, RGR2 or RGR3), β is vector of significant effects with incidence matrix **X**, $a \sim N(0, A\sigma^2_a)$ and $pe \sim N(0, I_c\sigma^2_{pe})$ are vectors of direct genetic and maternal permanent environmental effects with incidence matrices **Z**$_a$ and **Z**$_{pe}$, respectively, **m** $\sim$ **N(0**, $A\sigma^2_m$) is vector of maternal genetic effects with incidence matrix **Z**$_m$, and e $\sim$ N(0, **I**$_n$ $\sigma^2_e$) is a vector of random residual effects. $\sigma^2_a$, $\sigma^2_{pe}$, $\sigma^2_m$ and $\sigma^2_e$ represent additive genetic, maternal permanent environmental, maternal genetic, and residual variances, respectively. **A** is the additive genetic relationship matrix, while **I**$_c$ and **I**$_n$ are identity matrices of order equal to the number of ewes and the number of records, respectively. The convergence was assumed, when the change in maximum log-likelihood between the last two iterations was less than $10^{-4}$ [21].

Total heritability ($h^2_t$) was estimated according to [22] using the following formula.

$$h^2_t = \frac{\sigma^2_a + 0.5\sigma^2_m + 1.5\sigma_{am}}{\sigma^2_p}$$

(8)

Where $\sigma^2_a$, $\sigma^2_m$, $\sigma_{am}$ and $\sigma^2_p$ represent direct additive genetic variance, maternal genetic variance, covariance between direct and maternal additive genetic effects, and total phenotypic variance, respectively. Direct heritability ($h^2a$), maternal heritability ($h^2m$) and the proportion of maternal permanent environmental effects ($pe^2$) were calculated as the ratios of direct additive, maternal genetic, and maternal permanent environmental variances to the total phenotypic variance, respectively. The appropriate model for each trait was selected using the Log-likelihood ratio test (LRT), computed as twice the difference between the log-likelihoods of the full and reduced models. The LRT was assumed to follow a chi-square distribution with degree of freedom equal to the difference in the number of random covariance components between the two models [23]. An effect was considered significant if its inclusion caused a significant ($P < 0.05$) increase in log-likelihood compared to the model in which it was ignored. When the difference in log-likelihood did not exceed the critical chi-square value ($p < 0.05$), the simplest model with fewer parameters was selected as the best fit. Genetic trends were estimated by regressing the annual mean estimating breeding value (EBV) of the animals on their year of birth.

## Results

### Environmental effects

Table 1 presents a comprehensive overview of the descriptive statistics for growth rate and efficiency-related traits. Notably, the pedigree structure utilized in this study is sufficiently robust to yield accurate estimates of genetic parameters. The phenotypic coefficient of variation (CV) for the studied traits ranged from 15.88% (WT6) to 57.49% (GE2). Pre-weaning traits exhibited higher mean values than the corresponding post-weaning traits, and traits measured during the pre-weaning growth period had the lowest CVs compared to other growth periods. Least square means (± SE) for all studied traits are presented in Tables 2 and 3. Environmental factors, including birth year, birth season, and villages of CBBPs had a significant effect ($P < 0.001$) on all growth traits. On the other hand, lamb sex significantly influenced KR1, GE1, GE3, RGR1 and RGR3, while litter size at birth significantly affected WT6, KR1, GE3, RGR1 and RGR3 ($P < 0.05$). Ewe parity also had a significant effect on ADG1, ADG2, KR1, KR2, GE1 and RGR1 ($P < 0.05$).

### Genetic parameter estimates

Estimates of genetic parameters for the studied traits, obtained under the best fitting univariate models, are presented in Table 4. The model including only direct additive genetic effects as the unique known random effect was determined as the most appropriate model for all traits except WT6 and ADG1. The model including maternal genetic effects, without considering covariance between them (Model 3) was determined as the most suitable model for WT6 and ADG1. Direct heritability estimates ($h^2_a$) based on the selected models were 0.46±0.02, 0.44±0.02, 0.35±0.02, 0.21±0.02, 0.38±0.02,

**Table 2. Least-squares means ± S.E for daily gain and kleiber ratio.**

| Parameters | WT6 | ADG1 | ADG2 | ADG3 | KR1 | KR2 | KR3 |
|---|---|---|---|---|---|---|---|
| Birth year | *** | *** | *** | *** | *** | *** | *** |
| Village | *** | *** | *** | *** | *** | *** | *** |
| Dargegn | 13.93±0.08[b] | 76.55±0.74[b] | 49.51±1.14[b] | 44.60±1.25[b] | 14.00±0.07[b] | 6.82±0.16[c] | 4.45±0.13[b] |
| Molale | 14.74±0.08[a] | 66.16±0.72[d] | 70.60±1.11[a] | 53.63±1.17[a] | 12.45±0.07[e] | 9.22±0.15[a] | 5.55±0.12[a] |
| Sinamba | 11.66±0.08[e] | 62.930.72[e] | 47.84±1.11[b] | 40.75±1.18[c] | 12.81±0.07[c] | 7.71±0.15[b] | 4.41±0.12[b] |
| Tsehaysina | 12.37±0.15[c] | 80.87±1.31[a] | 35.18±2.06[d] | 18.69±3.97[d] | 12.74±0.13[d] | 4.97±0.28[e] | 2.37±0.40[c] |
| Zeram | 11.87±0.10[d] | 69.95±0.93[c] | 41.69±1.41[c] | 14.73±1.59[d] | 14.33±0.09[a] | 6.31±0.19[d] | 2.20±0.16[c] |
| Sex | ns | ns | ns | ns | ** | ns | ns |
| Male | 12.91±0.08 | 71.25±0.74 | 48.64+1.15 | 34.99+1.39 | 13.62+0.07 | 6.96+0.16 | 3.83+0.14 |
| Female | 12.91±0.08 | 71.33±0.74 | 49.29+1.15 | 33.97+1.38 | 13.71+0.07 | 7.05+0.16 | 3.76+0.14 |
| Birth type | ** | ns | ns | ns | * | ns | ns |
| Single | 12.75±0.04 | 70.90±0.39 | 47.16±0.60 | 36.42±0.93 | 13.54±0.04 | 6.80±0.08 | 3.98±0.09 |
| Twin | 13.08±0.15 | 71.69±1.31 | 50.77±2.03 | 32.54±2.25 | 13.80±0.13 | 7.21±0.28 | 3.61±0.23 |
| Parity | * | *** | ** | ns | *** | * | ns |
| 1 | 12.75±0.08[b] | 70.49±0.76[bc] | 49.14±1.17[ab] | 32.92±1.42 | 13.72±0.08[ac] | 7.12±0.16[ab] | 3.67±0.14 |
| 2 | 12.80±0.08[ab] | 69.71±0.76[bc] | 48.83±1.17[ab] | 33.83±1.41 | 13.55±0.08[bd] | 7.04±0.16[ab] | 3.78±0.14 |
| 3 | 12.83±0.08[ab] | 70.43±0.77[bc] | 47.97±1.17[b] | 33.32±1.43 | 13.59±0.08[bc] | 6.92±0.16[b] | 3.73±0.14 |
| 4 | 12.77±0.09[ab] | 69.41±0.79[bc] | 48.88±1.20[ab] | 34.42±1.46 | 13.48±0.08[bd] | 7.04±0.16[ab] | 3.76±0.15 |
| 5 | 12.91±0.09[a] | 68.72±0.84[c] | 51.26±1.29[a] | 33.19±1.55 | 13.39±0.08[d] | 7.33±0.18[a] | 3.65±0.16 |
| 6 | 12.87±0.10[ab] | 70.66±0.94[bc] | 48.33±1.44[ab] | 34.76±1.69 | 13.59±0.09[bcd] | 6.94±0.20[ab] | 3.81±0.17 |
| 7 | 13.06±0.12[a] | 71.24±1.15[bc] | 49.81±1.74[ab] | 35.97±1.91 | 13.64±0.11[abcd] | 7.07±0.24[ab] | 3.86±0.19 |
| 8 | 13.05±0.17[a] | 73.90±1.55[ab] | 48.33±2.44[ab] | 36.38±2.55 | 13.89±0.15[ab] | 6.82±0.34[ab] | 4.01±0.26 |
| >9 | 13.18±0.18[a] | 77.07±1.63[a] | 48.13±2.52[ab] | 35.53±2.55 | 14.13±0.16[a] | 6.78±0.35[ab] | 3.90±0.26 |
| Birth season | *** | *** | *** | *** | *** | *** | *** |
| Main rainy | 13.15±0.08[a] | 74.56±0.75[a] | 49.31±1.17[a] | 34.02±1.38[b] | 13.90±0.08[a] | 7.00±0.16[ab] | 3.77±0.14[b] |
| Dry | 12.55±0.08[b] | 69.38±0.74[b] | 46.64±1.15[b] | 36.89±1.39[a] | 13.48±0.07[c] | 6.82±0.16[b] | 3.99±0.14[a] |
| Short rainy | 13.04±0.09[a] | 69.93±0.81[b] | 50.95±1.25[a] | 32.54±1.53[b] | 13.61±0.08[b] | 7.20±0.17[a] | 3.62±0.15[b] |

[abcde]On the same column, numbers bearing the same superscript are not statistically different at P=0.05-P>0.05. ns: not significant. ***P<0.001; ** P<0.01 and * P<0.05; WT6: six months weight, ADG1: daily gain from birth to weaning, ADG2: daily gain from weaning to six months, ADG3: daily gain from six months to yearling, KR1: kleiber ratio from birth to weaning, KR2: kleiber ratio from weaning to six months, KR3: kleiber ratio from six months to yearling.

0.36±0.02, 0.23±0.02, 0.39±0.02, 0.39±0.02, 0.48±0.03, 0.36±0.02, 0.42±0.02, and 0.47±0.03 for WT6, ADG1, ADG2, ADG3, KR1, KR2, KR3, GE1, GE2, GE3, RGR1, RGR2, and RGR3, respectively. Maternal heritability ($h^2_m$) for WT6 and ADG1 was 0.02±0.01 and 0.02±0.01. Estimates of additive coefficient of variation varied from 9.35% in KR1 to 34.21% in GE2.

## Phenotypic and genetic correlations

Estimates of phenotypic and genetic correlations (± SE) from multivariate analysis are presented in Table 5. Genetic correlations raged from −0.81±0.03 for KR1-KR2 to 0.98±0.01 for KR2-GE2, KR2-RGR2 and GE1-RGR1. Phenotypic correlations varied from −0.73±0.01 for KR1-KR2 to 0.98±0.001 for KR2-RGR2 and GE2-RGR2. Furthermore, genetic and phenotypic correlations between WT6 and traits measured during the pre-weaning and weaning to six months growth periods were positive, with magnitudes ranging from low to moderate.

**Table 3. Least-squares means ± S.E for growth efficiency and relative growth rate.**

| Parameters | GE1 | GE2 | GE3 | RGR1 | RGR2 | RGR3 |
|---|---|---|---|---|---|---|
| Birth year | *** | *** | *** | *** | *** | *** |
| Village | *** | *** | *** | *** | *** | *** |
| Dargegn | 256.45 ± 3.322[c] | 47.15 ± 1.37[c] | 57.59 ± 1.49[b] | 1.40 ± 0.01[b] | 0.42 ± 0.01[c] | 0.25 ± 0.005[b] |
| Molale | 224.42 ± 3.14[e] | 77.63 ± 1.34[a] | 38.81 ± 1.45[c] | 1.23 ± 0.01[d] | 0.59 ± 0.01[a] | 0.18 ± 0.005[c] |
| Sinamba | 250.09 ± 3.14[d] | 52.92 ± 1.34[b] | 68.40 ± 1.42[a] | 1.33 ± 0.01[c] | 0.45 ± 0.01[b] | 0.28 ± 0.005[a] |
| Tsehaysina | 313.44 ± 5.70[b] | 23.64 ± 2.47[e] | 36.33 ± 4.34[d] | 1.56 ± 0.02[a] | 0.23 ± 0.02[e] | 0.17 ± 0.015[cd] |
| Zeram | 330.20 ± 4.03[a] | 42.95 ± 1.69[d] | 34.02 ± 1.80[d] | 1.60 ± 0.01[a] | 0.39 ± 0.01[d] | 0.16 ± 0.006[d] |
| Sex | *** | ns | * | *** | ns | * |
| Male | 271.99 ± 3.22 | 48.41 ± 1.38 | 47.73 ± 1.57 | 1.42 ± 0.01 | 0.42 ± 0.01 | 0.21 ± 0.01 |
| Female | 277.85 ± 3.23 | 49.31 ± 1.38 | 46.33 ± 1.56 | 1.44 ± 0.01 | 0.42 ± 0.01 | 0.20 ± 0.01 |
| Birth type | ns | ns | * | * | ns | * |
| Single | 270.56 ± 1.68 | 49.37 ± 0.71 | 49.65 ± 1.03 | 1.41 ± 0.01 | 0.42 ± 0.01 | 0.21 ± 0.003 |
| Twin | 279.28 ± 5.72 | 48.34 ± 2.45 | 44.41 ± 2.57 | 1.45 ± 0.02 | 0.42 ± 0.02 | 0.20 ± 0.01 |
| Parity | * | ns | ns | *** | ns | ns |
| 1 | 282.00 ± 3.31[a] | 48.95 ± 1.41 | 47.44 ± 1.60 | 1.45 ± 0.01[a] | 0.42 ± 0.01 | 0.21 ± 0.01 |
| 2 | 272.03 ± 3.31[b] | 49.30 ± 1.41 | 47.03 ± 1.60 | 1.42 ± 0.01[b] | 0.42 ± 0.01 | 0.20 ± 0.01 |
| 3 | 273.20 ± 3.33[b] | 48.22 ± 1.41 | 45.21 ± 1.61 | 1.42 ± 0.01[b] | 0.42 ± 0.01 | 0.20 ± 0.01 |
| 4 | 268.39 ± 3.42[bc] | 49.22 ± 1.45 | 47.78 ± 1.64 | 1.41 ± 0.01[bc] | 0.42 ± 0.01 | 0.21 ± 0.01 |
| 5 | 263.49 ± 3.67[c] | 51.70 ± 1.55 | 47.00 ± 1.76 | 1.39 ± 0.01[c] | 0.44 ± 0.01 | 0.21 ± 0.01 |
| 6 | 270.63 ± 4.10[b] | 49.29 ± 1.73 | 48.61 ± 1.93 | 1.41 ± 0.01[bc] | 0.42 ± 0.01 | 0.21 ± 0.01 |
| 7 | 273.19 ± 5.01[b] | 50.19 ± 2.07 | 46.50 ± 2.21 | 1.42 ± 0.02[b] | 0.43 ± 0.01 | 0.20 ± 0.01 |
| 8 | 282.49 ± 6.73[a] | 48.26 ± 2.91 | 46.98 ± 3.03 | 1.45 ± 0.02[a] | 0.41 ± 0.02 | 0.21 ± 0.01 |
| >9 | 288.87 ± 7.09[a] | 44.59 ± 3.02 | 46.74 ± 3.09 | 1.47 ± 0.02[a] | 0.39 ± 0.02 | 0.21 ± 0.01 |
| Birth season | *** | *** | *** | *** | *** | *** |
| Main rainy | 282.61 ± 3.28[a] | 46.97 ± 1.40[b] | 44.37 ± 1.57[b] | 1.45 ± 0.01[a] | 0.40 ± 0.01[b] | 0.20 ± 0.01[b] |
| Dry | 267.70 ± 3.23[c] | 48.05 ± 1.38[b] | 51.74 ± 1.58[a] | 1.40 ± 0.01[c] | 0.41 ± 0.01[b] | 0.22 ± 0.01[a] |
| Short rainy | 274.46 ± 3.53[b] | 51.55 ± 1.50[a] | 44.98 ± 1.73[b] | 1.43 ± 0.01[b] | 0.44 ± 0.01[a] | 0.20 ± 0.01[b] |

[abcde]On the same column, numbers bearing the same superscript are not statistically different at P = 0.05-P > 0.05. ns: not significant. ***P < 0.001; **P < 0.01 and *P < 0.05; GE1: growth efficiency from birth to weaning, GE2: growth efficiency from weaning to six months, GE3: growth efficiency from six months to yearling, RGR1: relative growth rate from birth to weaning, RGR2: relative growth rate from weaning to six months, RGR3: relative growth rate from six months to yearling.

## Genetic trends

Annual genetic trends for six-month weight by year of birth are shown in Fig 1, with WT6 showing an improvement at a rate of 0.0384 kg. Fig 2 illustrates the genetic trends for average daily gain and growth efficiency from weaning to six months of age, with annual improvement of 0.2964 g and 0.4968, respectively. Genetic trends for Kleiber ratio and relative growth rate over the same period are shown in Fig 3, exhibiting positive improvement at a rate of 0.0499 and 0.0035, respectively. Traits that did not exhibit meaningful genetic changes are not presented in the figures.

## Discussion

Birth year was a significant environmental factor for all studied traits. The effect of year on growth rate and efficiency-related traits likely reflects difference in climatic conditions, the indirect effect of selective breeding for 6-months weight, and variations in nutrition and management practices. Several studies have also reported a significant effect of birth

**Table 4. Estimate of variance components and heritability for efficiency-related traits.**

| Traits | M | $\sigma^2_a$ | $\sigma^2_m$ | $\sigma_{am}$ | $\sigma^2_{pe}$ | $\sigma^2_e$ | $\sigma^2_P$ | $h^2_a$ | $h^2_m$ | $r_{am}$ | $pe^2$ | $h^2_t$ | CVA (%) | Log (L) |
|---|---|---|---|---|---|---|---|---|---|---|---|---|---|---|
| WT6 | 3 | 1.691 | 0.089 | | | 1.871 | 3.651 | 0.46±0.02 | 0.02±0.01 | | | 0.47 | 10.36 | −17338.94 |
| ADG1 | 3 | 162.444 | 7.496 | | | 200.47 | 370.41 | 0.44±0.02 | 0.02±0.01 | | | 0.45 | 19.33 | −65039.59 |
| ADG2 | 1 | 248.909 | | | | 460.068 | 708.977 | 0.35±0.02 | | | | 0.35±0.02 | 31.6 | −56790.49 |
| ADG3 | 1 | 106.271 | | | | 406.622 | 512.893 | 0.21±0.02 | | | | 0.21±0.02 | 23.23 | −32397.02 |
| KR1 | 1 | 1.46 | | | | 2.349 | 3.809 | 0.38±0.02 | | | | 0.38±0.02 | 9.35 | −21696.14 |
| KR2 | 1 | 4.935 | | | | 8.75 | 13.865 | 0.36±0.02 | | | | 0.36±0.02 | 29.9 | −26970.62 |
| KR3 | 1 | 1.242 | | | | 4.048 | 5.29 | 0.23±0.02 | | | | 0.23±0.02 | 23.67 | −11902.12 |
| GE1 | 1 | 2964.83 | | | | 4604.68 | 7569.51 | 0.39±0.02 | | | | 0.39±0.02 | 22.23 | −93938.51 |
| GE2 | 1 | 369.644 | | | | 565.528 | 935.172 | 0.39±0.02 | | | | 0.39±0.02 | 34.21 | −56782.92 |
| GE3 | 1 | 305.224 | | | | 329.595 | 634.82 | 0.48±0.03 | | | | 0.48±0.03 | 30.17 | −30225.13 |
| RGR1 | 1 | 0.025 | | | | 0.044 | 0.069 | 0.36±0.02 | | | | 0.36±0.02 | 11.98 | 16342.13 |
| RGR2 | 1 | 0.017 | | | | 0.024 | 0.041 | 0.42±0.02 | | | | 0.42±0.02 | 27.74 | 16393.71 |
| RGR3 | 1 | 0.003 | | | | 0.004 | 0.007 | 0.47±0.03 | | | | 0.47±0.03 | 22.82 | 16130.19 |

$\sigma^2_P$ phenotypic variance; $\sigma^2_a$ additive variance; $\sigma^2_m$ maternal variance; $\sigma^2_{pe}$ common environment variance; $\sigma^2_e$ error variance; $h^2_a$ direct heritability; $h^2_m$ maternal heritability; $h^2_{pe}$ ration of common environment variance to the total phenotypic variance; $h^2_t$ total heritability; $r_{am}$ genetic correlation between direct and maternal additive heritability; $\sigma_{am}$.covariance between direct and maternal additive genetic effect; $CV_A$ additive coefficient of variance; Log (L) log Likelihood. WT6: six months weight, ADG1: daily gain from birth to weaning, ADG2: daily gain from weaning to six months, ADG3: daily gain from six months to yearling, KR1: kleiber ratio from birth to weaning, KR2: kleiber ratio from weaning to six months, KR3: kleiber ratio from six months to yearling, GE1: growth efficiency from birth to weaning, GE2: growth efficiency from weaning to six months, GE3: growth efficiency from six months to yearling, RGR1: relative growth rate from birth to weaning, RGR2: relative growth rate from weaning to six months, RGR3: relative growth rate from six months to yearling.

year on growth rate and efficiency-related traits in different sheep breeds [18,24–31]. All studied traits were significantly influenced by CBBPs villages. Differences among villages likely reflect variations in flock management practice, climatic conditions, ram selection intensity, availability of grazing pasture in quantity and quality and other unknown environmental factors. The observed differences between male and female lambs in growth and efficiency-related traits can be attributed to sex-specific hormonal effects, with testosterone promoting growth in male and estrogen limiting the elongation of long bones in females [18,26]. Female lambs exhibited higher growth rates and greater feed conversion efficiency than males during the pre-weaning growth period. However, male lambs, tended to surpass females in growth rate from six months to yearling age. In contrast to the present studies, [24–26,28] reported that male lambs showed superior growth rate and efficiency related traits across all growth periods. Twin-born lambs demonstrated higher feed utilization and relative growth rates than singletons during pre-weaning period. On the other hand, singleton exhibited higher growth efficiency and relative growth rate from six months to yearling age, which contrasts with findings reported for Kermani sheep [24] and Baluchi sheep [18]. This discrepancy may be due to the relatively small number of twin-born lambs in our study. The significant effect of parity on the studied traits align with previous reports [18,24,25,28], and likely reflects difference in maternal effects, including nursing and mothering ability of the dam at different ages. Seasonal variations in feed availability, both in quality and quantity, as well as disease incidence, may also contribute to these effects. Similar effects of birth seasons on growth and efficiency traits have been reported in various sheep breeds [28–31] in different sheep breeds.

The estimated (co)variance components and corresponding genetic parameters obtained from the appropriate models are presented in Table 4. Based on the log-likelihood values, Model 1 was identified as the most suitable model for all traits except WT6 and ADG1, Indicating that accounting solely for direct genetic effects as the unique random effect provided the best fit for these traits. In contrast, Model 3, which incorporated both direct and maternal genetic effects without considering their covariance, was determined to be the most appropriate model for WT6 and ADG1. This finding highlights the importance of maternal genetic effects in the genetic analysis of this trait in Menz sheep. Consistent with the present

Table 5. Genetic (above diagonal) and Phenotypic (below diagonal) correlation.

| Traits | WT6 | ADG1 | ADG2 | ADG3 | KR1 | KR2 | KR3 | GE1 | GE2 | GE3 | RGR1 | RGR2 | RGR3 |
|---|---|---|---|---|---|---|---|---|---|---|---|---|---|
| WT6 |  | 0.40±0.06 | 0.40±0.06 | -0.12±0.09 | 0.31±0.06 | 0.17±0.08 | -0.37±0.08 | 0.27±0.06 | 0.37±0.06 | -0.50±0.06 | 0.21±0.06 | 0.34±0.06 | -0.49±0.06 |
| ADG1 | 0.30±0.01 |  | -0.67±0.10 | 0.05±0.10 | 0.96±0.01 | -0.69±0.04 | -0.11±0.08 | 0.89±0.02 | -0.68±0.04 | -0.13±0.08 | 0.85±0.02 | -0.58±0.04 | -0.04±0.07 |
| ADG2 | 0.47±0.01 | -0.68±0.01 |  | -0.15±0.11 | -0.58±0.05 | 0.97±0.01 | -0.37±0.09 | -0.57±0.06 | 0.96±0.01 | -0.26±0.09 | -0.59±0.06 | 0.96±0.01 | -0.32±0.09 |
| ADG3 | -0.31±0.02 | 0.02±0.02 | -0.25±0.02 |  | -0.01±0.09 | -0.22±0.10 | 0.96±0.01 | 0.12±0.09 | -0.23±0.10 | 0.91±0.02 | -0.10±0.07 | -0.16±0.08 | 0.88±0.09 |
| KR1 | 0.25±0.01 | 0.95±0.001 | -0.54±0.01 | -0.02±0.02 |  | -0.81±0.03 | -0.05±0.09 | 0.96±0.01 | -0.74±0.04 | -0.09±0.08 | 0.96±0.01 | -0.66±0.04 | -0.19±0.07 |
| KR2 | 0.33±0.01 | -0.68±0.01 | 0.97±0.001 | -0.28±0.02 | -0.73±0.01 |  | -0.15±0.11 | -0.71±0.05 | 0.98±0.01 | -0.14±0.10 | -0.57±0.04 | 0.98±0.01 | -0.26±0.08 |
| KR3 | -0.54±0.01 | -0.07±0.02 | -0.45±0.01 | 0.97±0.01 | -0.04±0.02 | -0.29±0.02 |  | 0.02±0.10 | -0.21±0.11 | -0.14±0.07 | -0.14±0.07 | -0.23±0.07 | 0.93±0.09 |
| GE1 | 0.33±0.01 | 0.86±0.004 | -0.45±0.01 | -0.01±0.02 | 0.90±0.002 | -0.56±0.01 | -0.03±0.02 |  | -0.57±0.06 | 0.09±0.08 | 0.98±0.004 | -0.54±0.06 | 0.23±0.09 |
| GE2 | 0.34±0.01 | -0.68±0.01 | 0.95±0.001 | -0.22±0.02 | -0.70±0.01 | 0.97±0.001 | -0.27±0.02 | -0.41±0.01 |  | -0.45±0.07 | -0.58±0.04 | 0.97±0.01 | -0.25±0.08 |
| GE3 | -0.55±0.01 | -0.12±0.02 | -0.33±0.02 | 0.93±0.002 | -0.10±0.02 | -0.29±0.02 | 0.97±0.001 | -0.11±0.02 | -0.32±0.02 |  | -0.17±0.07 | -0.24±0.07 | 0.93±0.09 |
| RGR1 | 0.31±0.01 | 0.84±0.004 | -0.47±0.01 | -0.04±0.02 | 0.96±0.001 | -0.57±0.01 | -0.06±0.02 | 0.96±0.001 | -0.58±0.01 | -0.10±0.02 |  | -0.47±0.01 | -0.10±0.02 |
| RGR2 | 0.35±0.01 | -0.58±0.01 | 0.97±0.001 | -0.26±0.02 | -0.68±0.01 | 0.98±0.001 | -0.34±0.02 | -0.43±0.01 | 0.98±0.001 | -0.34±0.02 | -0.47±0.01 |  | -0.33±0.02 |
| RGR3 | -0.56±0.01 | -0.19±0.02 | -0.32±0.02 | 0.90±0.02 | -0.10±0.02 | -0.37±0.02 | 0.94±0.02 | -0.08±0.02 | -0.34±0.02 | 0.93±0.02 | -0.10±0.02 | -0.33±0.02 |  |

WT6: six month weight; ADG1: average daily gain from birth to weaning, ADG2: average daily gain from weaning to six months, ADG3: average daily gain from six months to yearling, KR1: kleiber ratio from birth to weaning, KR2: kleiber ratio from weaning to six months, KR3: kleiber ratio from six months to yearling, GE1: growth efficiency from birth to weaning, GE2: growth efficiency from weaning to six months, GE3: growth efficiency from six months to yearling, RGR1: relative growth rate from birth to weaning, RGR2: relative growth rate from weaning to six months, RGR3: relative growth rate from six months to yearling.

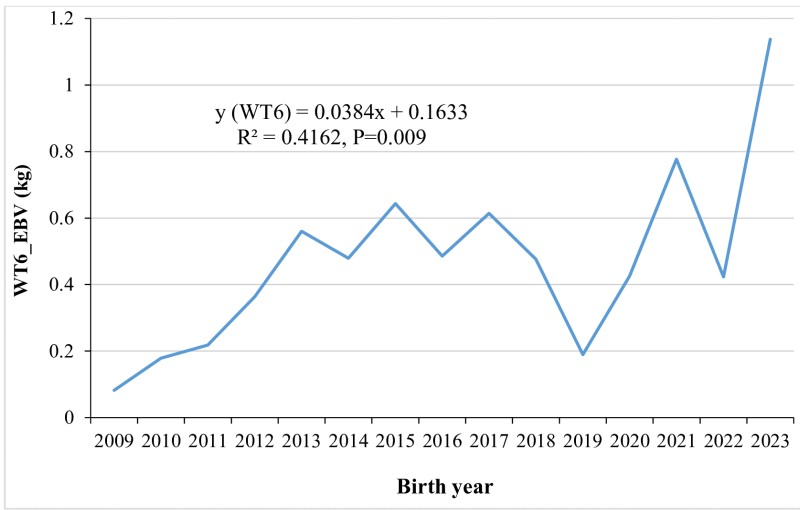

**Fig 1. Genetic trends for six months weight by year of birth.**

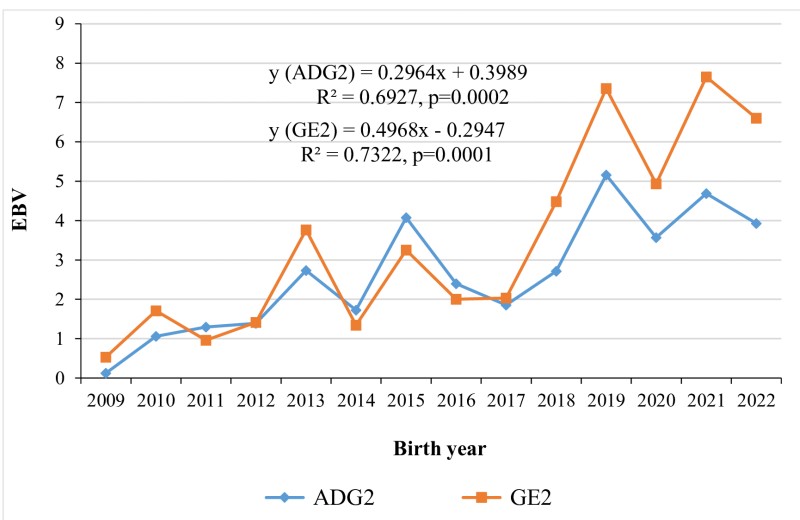

**Fig 2. Genetic trends for average daily gain (g) and growth efficiency from weaning to six months of age.**

study, [18] reported as similar model for all traits except KR1, GE1, and RGR1 in the Baluchi sheep. In contrast, [24] recommended different models that included both maternal and permanent environmental effects in Kermani sheep. Based on the most suitable models identified, direct heritability estimates for the studied traits ranged from $0.21 \pm 0.02$ (ADG3) to $0.48 \pm 0.03$ (GE3). Genetic parameter estimates for efficiency-related traits remain scarce in the literature. The present direct heritability estimates for growth rate and efficiency-related traits were higher than those reported for Dorper sheep [30], Kermani sheep [24], and Baluchi sheep [18]. In general, direct heritability estimates for growth rate and efficiency related traits in different sheep breeds reported in the literature ranged from 0.00 to 0.45 [18,24,25,28–34]. Variation in sheep breed, genetic diversity among populations, differences in data structure, and the methods applied for genetic parameter estimation are potential source of discrepancies between the present findings and those reported for other

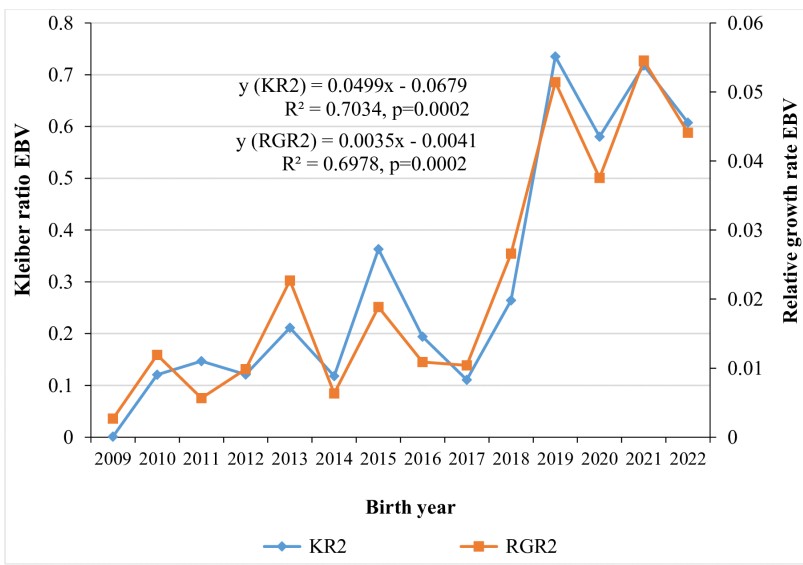

**Fig 3. Genetic trends for kleiber ratio and relative growth rate from weaning to six months of age.**

sheep breeds [32]. The moderate to high direct heritability estimates observed in this study suggest the presence of substantial additive genetic variation for the studied traits. Consequently, considerable genetic progress would be achieved through direct genetic selection.

Estimates of genetic and phenotypic correlation between the studied traits are presented in Table 5. The strong and positive direct genetic correlations observed among traits within the same growth periods suggest that pleiotropic genes may underline these associations [24,26]. Consequently, genetic selection to improve one trait within the same growth period is expected to result in favorable correlated response in the others. Consistent with our results, [26] reported similarly strong and positive genetic correlations among growth traits in Raeini Cashmere goats. The genetic correlations between ADG1-KR1, ADG2-KR2, and ADG3-KR3 were $0.96 \pm 0.01$, $0.97 \pm 0.01$, and $0.96 \pm 0.01$, respectively; indicating that lambs with higher growth rates are also more efficient feed utilizers. Moreover, these results indicate that selection for pre-weaning growth rate increases feed utilization efficiency during the same growth period, but slightly reduce efficiency in the post-weaning period. Comparable trends have been reported in Zandi sheep [15] and Kermani lambs [24].

Phenotypic correlations among the studied traits within the same growth period ranged from $0.84 \pm 0.004$ for ADG1-RGR1 to $0.98 \pm 0.001$ for KR2-RGR2, GE2-RGR2. The strong and positive phenotypic correlations observed in the present study suggest that selection for any one of these traits would result in favourable phenotypic response in the others within the same growth period. In contrast, genetic and phenotypic correlations between pre and post-weaning growth periods were unfavourable, revealed that lambs with higher growth rate and efficiency-related traits during the pre-weaning periods tend to perform less efficiently post-weaning, and vice versa. This revealed that different genetic mechanisms may underline the expression of these traits across growth periods [35]. On the other hand, genetic and phenotypic correlations between WT6 and traits measured during pre-weaning and weaning to six months growth periods were positive but of low to moderate magnitude. Therefore, continuing selection program under the current selection criteria could lead to improvements in efficiency-related traits. Although growth rate and efficiency-related traits were not direct selection objectives in Menz community-based breeding programs, the observed annual genetic improvements in these traits are likely attributed to correlated response (Figs 1-3).

## Conclusion

The results of this study indicate that growth rate and efficiency-related traits were significantly influenced by year of birth, season, CBBPs villages, and other environmental factors. Therefore, these effects should be accounted for when estimating genetic parameter for these traits. Direct additive genetic effects as the unique known random effect provided the best fit for all traits, except pre-weaning average daily gain (ADG1), for which a model including both direct and maternal genetic effects was more appropriate. Moderate to high direct heritability estimates and additive genetic coefficient of variation suggest that direct selection could achieve substantial genetic progress for these traits. Traits measured within the same growth period exhibited strong and positive genetic and phenotypic correlations. Given that, Menz sheep are reared under harsh environmental conditions, incorporating efficiency-related traits into the selection program could improve overall breeding program efficiency. Selection for WT6 would be more effective and improve efficiency related traits due to the observed positive genetic correlations.

## Supporting information

**S1 Data. Body weight and growth efficiency-related data.**
(XLSX)

## Acknowledgments

The authors sincerely acknowledge the international Center for Agricultural Research in the Dry Ares (ICARDA) for their invaluable assistance and provision of technical expertise and materials support in data analysis and running breeding programs. We also thanks the enumerators who over the years have collected the data and managing the breeding programs.

## Author contributions

**Conceptualization:** Asfaw Bisrat, Shanbel Besufkad, Aschalew Abebe, Shenkute Goshme, Ayele Abebe, Tesfaye Zewdie, Alemnew Areaya, Zeleke Tesema, Chekol Demis, Erdachew Yitagesu, Yeshitila Wondifra, Tadiwos Asfaw, Enyiew Alemnew, Solomon Gizaw, Tesfaye Getachew, Barbara Rischkowsky, Mourad Rekik, Berhanu Belay, Aynalem Haile.

**Data curation:** Asfaw Bisrat, Shanbel Besufkad, Aschalew Abebe, Shenkute Goshme, Ayele Abebe, Tesfaye Zewdie, Alemnew Areaya, Zeleke Tesema, Chekol Demis, Erdachew Yitagesu, Yeshitila Wondifra, Tadiwos Asfaw, Enyiew Alemnew, Solomon Gizaw, Tesfaye Getachew, Barbara Rischkowsky, Mourad Rekik, Berhanu Belay, Aynalem Haile.

**Formal analysis:** Asfaw Bisrat, Shanbel Besufkad.

**Investigation:** Aschalew Abebe, Shenkute Goshme, Ayele Abebe, Tesfaye Zewdie, Alemnew Areaya, Zeleke Tesema, Chekol Demis, Yeshitila Wondifra, Tadiwos Asfaw, Enyiew Alemnew, Solomon Gizaw, Tesfaye Getachew, Barbara Rischkowsky, Mourad Rekik, Berhanu Belay, Aynalem Haile.

**Methodology:** Asfaw Bisrat, Shanbel Besufkad, Aschalew Abebe, Shenkute Goshme, Ayele Abebe, Tesfaye Zewdie, Alemnew Areaya, Zeleke Tesema, Chekol Demis, Erdachew Yitagesu, Yeshitila Wondifra, Tadiwos Asfaw, Enyiew Alemnew, Solomon Gizaw, Tesfaye Getachew, Barbara Rischkowsky, Mourad Rekik, Berhanu Belay, Aynalem Haile.

**Project administration:** Erdachew Yitagesu.

**Software:** Shanbel Besufkad.

**Supervision:** Erdachew Yitagesu, Yeshitila Wondifra, Tadiwos Asfaw, Enyiew Alemnew, Solomon Gizaw, Tesfaye Getachew, Barbara Rischkowsky, Mourad Rekik, Berhanu Belay, Aynalem Haile.

**Validation:** Aschalew Abebe, Shenkute Goshme, Ayele Abebe, Tesfaye Zewdie, Alemnew Areaya, Zeleke Tesema, Chekol Demis, Erdachew Yitagesu, Yeshitila Wondifra, Tadiwos Asfaw, Enyiew Alemnew, Solomon Gizaw, Tesfaye Getachew, Barbara Rischkowsky, Mourad Rekik, Berhanu Belay, Aynalem Haile.

**Visualization:** Aschalew Abebe, Shenkute Goshme, Ayele Abebe, Tesfaye Zewdie, Alemnew Areaya, Zeleke Tesema, Chekol Demis, Solomon Gizaw, Tesfaye Getachew, Barbara Rischkowsky, Mourad Rekik, Berhanu Belay, Aynalem Haile.

**Writing – original draft:** Asfaw Bisrat, Shanbel Besufkad.

**Writing – review & editing:** Asfaw Bisrat, Shanbel Besufkad, Solomon Gizaw, Tesfaye Getachew.

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
