## [Decision Letter · Decision Letter 0]

18 Sep 2025

Dear Dr. Shanbel Besufkad,

Thank you for submitting your manuscript to PLOS ONE. After careful consideration, we feel that it has merit but does not fully meet PLOS ONE’s publication criteria as it currently stands. Therefore, we invite you to submit a revised version of the manuscript that addresses the points raised during the review process.

We look forward to receiving your revised manuscript.

Kind regards,

Nesrein M. Hashem

Academic Editor

PLOS ONE

Journal Requirements:

https://sanad.iau.ir/journal/ijas/Article/665354_1132664?jid=665354_1132664&_action=article&_kw=Kleiber+ratio&kw=122655

https://linkinghub.elsevier.com/retrieve/pii/S2405844024010028

In your revision ensure you cite all your sources (including your own works), and quote or rephrase any duplicated text outside the methods section. Further consideration is dependent on these concerns being addressed.

“The Menz sheep community-based breeding program was financially supported by CGIAR initiative on “Sustainable Animal Productivity for Livelihoods, Nutrition and Gender inclusion (SAPLING),” Accelerating the Impact of CGIAR Climate Research in Africa (AICCRA) project, Amhara Agricultural Research Institute (ARARI) and Ethiopian Institute of Agricultural Research (EIAR).”

6. Please include a separate caption for each figure in your manuscript.

Reviewers' comments:

Reviewer's Responses to Questions

**Comments to the Author**

1. Is the manuscript technically sound, and do the data support the conclusions?

Reviewer #1: Yes

Reviewer #2: Yes

2. Has the statistical analysis been performed appropriately and rigorously?

Reviewer #1: Yes

Reviewer #2: Yes

3. Have the authors made all data underlying the findings in their manuscript fully available?

Reviewer #1: Yes

Reviewer #2: No

4. Is the manuscript presented in an intelligible fashion and written in standard English?

Reviewer #1: No

Reviewer #2: No

Reviewer #1: There are a lot of typographical errors. Please look in to it before final resubmission. All the grammatical errors and sentence corrections must be done.

Results section has one heading: environmental effects. All of your results are given under this which gives a wrong impression. Please edit accordingly.

Reviewer #2: Thanks for inviting manuscript number PONE-D-25-45437: Genetic evaluation of growth rate and efficiency-related traits in Menz sheep. I try to review comments, and I will give detailed below

Title

consider shortening to “Genetic parameters of growth and efficiency traits in Menz sheep” for conciseness. I also saw an article on Dorper at the same research center by same author

• “Efficiency-related traits” could be specified (e.g., Kleiber ratio, relative growth rate). Kleiber is not commonly used. Put research gap

Abstract

• L14–16: The phrase “traits related to growth efficiency are economically important and genetic improvement of these traits is vital” is slightly repetitive. Suggest: “Growth efficiency traits are economically important, and their genetic improvement is essential.”

• L17–22: The abstract lists too many traits (6MW, ADG1, ADG2, ADG3, KR1, etc.) without context. This overwhelms the reader. Consider grouping (e.g., “average daily gain, Kleiber ratios, efficiency of growth, and relative growth rate”).

• L29–32: “Direct additive genetic effect as the unique known random effect was the best fitting model …” → grammatically awkward. Suggest: “The model with direct additive genetic effect alone provided the best fit for most traits, except 6MW and ADG1, where maternal effects were also important.”

• L33–37: Heritability estimates are listed as a long series of numbers. Better to summarize ranges (e.g., “heritability ranged from 0.21 to 0.48 across traits”).

• L42–44: “Selection for 6MW would be more effective …” → Please clarify why six-month weight is the most informative trait. Already published so many times, repeated finding. Think new idea, a new suggestion that can advance the ongoing CBBP

Keywords

• Good selection, but you may add “Menz sheep” explicitly as a keyword to improve discoverability.

Introduction

• L49–56: Good background, but sentence “Small ruminants have been significantly supporting the livelihoods of smallholder farmers…” is awkward. Suggest: “Small ruminants play a major role in supporting the livelihoods of smallholder farmers across Ethiopia’s diverse ecological zones.”

• L57–66: The problem statement (low productivity despite adaptation) is strong. However, you need to clearly define the research gap: Few studies have estimated genetic parameters for efficiency traits (Kleiber ratio, RGR, GE) in Menz sheep.

• L67–72: The introduction to Kleiber ratio is appropriate, but the rationale for including efficiency traits should be emphasized. Add supporting references from similar indigenous sheep studies.

• Kleiber? I did not understand its importance

• L73–78: The objective statement is clear, but could be sharper: “This study aimed to estimate genetic parameters for growth rate and efficiency-related traits in Menz sheep under community-based breeding programs.”

Materials and Methods

• Location & flock description (L80–96):

o Good detail about geography and climate. However, some redundancy (adaptability of Menz sheep already mentioned in Introduction).

o Suggest summarizing climatic info in fewer words, and moving references on adaptability to Introduction.

• Data collection (L97–111):

o L97: “New born lambs were weighted” → “weighed.”

o L105: “…primarily focused on using six months weight as selection criteria” → “as the main selection criterion.” Since already described by other author just cite the previous authors. The procedure of CBBP selection is as described by ----------

o L109–111: Provide clarity: “Rams were rotated annually between groups to minimize inbreeding.” Still duplication

• Dataset (L112–126):

o Excellent that you specify sample size (22,712 records, 6,528 ewes).

o L117–118: Typo: ADG1 repeated twice (birth–weaning and weaning–6 months). One should be ADG2.

• Trait definitions (L127–137):

o Equations should be formatted clearly with subscripts and consistent notation.

o L133: Efficiency of growth formula is confusing; add units or explanation (percentage change relative to initial weight).

• Statistical models (L138–192):

o L172: “…when the change in maximum log L between the last two iterations is less than 10-4” → please write as 1e-4 for clarity.

o Genetic trend analysis (L190–192): Provide more detail on the time scale (2009–2023). Were all traits analyzed for trends or only key ones? Need clarification

Results

• L193–207: Environmental effects are described well. However:

o Phrase “pedigree structures utilized in this study are fair enough to accurate genetic parameters” → awkward. Suggest: “The pedigree structure was adequate for accurate estimation of genetic parameters.”

• L208–217: Heritability values should again be summarized in ranges, with key examples. The full list can stay in the table.

• L218–229: Correlations are strong, but the text is hard to follow. Suggest breaking into sub-sections: (i) within-period correlations, (ii) across-period correlations.

• Figures 1–3: Ensure captions explain the units of genetic trend (kg/year, g/day/year, etc.).

Discussion

• L231–239: Good explanation of environmental effects. Could improve by citing more Ethiopian sheep literature beyond Baluchi/Kermani references.

• L243–246: The hormonal explanation for sex differences is good, but speculative. Please cite a specific physiological reference.

• L248–252: The finding that twins were more efficient pre-weaning but less efficient post-weaning is very interesting. This needs stronger discussion and reference support.

• L260–283: Discussion on heritability estimates is strong. However, some sentences are too long and repetitive. Simplify.

• L285–307: The point about unfavorable pre- vs. post-weaning correlations is very important. Please emphasize its implication: selection should consider multi-trait indices to avoid negative correlated responses.

Conclusion

• L315–329: Well-structured, but slightly repetitive. Suggest trimming.

• Add practical recommendation: e.g., “Breeding programs should prioritize six-month weight while monitoring efficiency-related traits to ensure balanced progress.” Too many times documented- see previous article including Bonga, Doyogena

References

• Generally good coverage, including Ethiopian studies.

• Some inconsistencies:

o L366: “and a pastoral system of ethiopia” → capitalize Ethiopia.

o Several references (e.g., #5) are poorly formatted and need editing.

• Please ensure consistent journal abbreviations (Small Ruminant Res. vs. Small Rumin Res.).

Tables & Figures

• Tables are informative but crowded. Suggest splitting if possible.

• Use consistent superscript letters for multiple comparisons (currently a, b, c, but sometimes overlapping).

• Figures of genetic trends should include error bars or confidence intervals.

**Do you want your identity to be public for this peer review?** For information about this choice, including consent withdrawal, please see our Privacy Policy

Reviewer #1: No

Reviewer #2: No

---

## [Author Response · Author response to Decision Letter 1]

30 Sep 2025

Responses to reviewer comments and questions

Dear editors and reviewers;

Thank you very much for taking the time to review our manuscript entitled "Genetic evaluation of growth rate and efficiency-related traits in Menz sheep” (PONE-D-25-45437). We greatly appreciate your insightful comments and constructive suggestions. Please find below our detailed responses to your feedback, along with the revised version of the manuscript, which has been resubmitted for your consideration.

Corresponding author ORCID: https://orcid.org/0000-0002-8984-9718

Reviewer point #1: There are a lot of typographical errors. Please look in to it before final resubmission. All the grammatical errors and sentence corrections must be done.

Author response #1: We greatly thank the reviewer for pointing this out. We have carefully reviewed the manuscript and corrected all typographical and grammatical errors. Additionally, sentences have been revised throughout to improve clarity, readability, and overall quality of the manuscript.

Reviewer point #2: Results section has one heading: environmental effects. All of your results are given under this which gives a wrong impression. Please edit accordingly.

Author response #2: We thank the reviewer for this observation. We have revised the Results section by reorganizing the headings to accurately reflect the content, ensuring that each set of results is presented under appropriate and distinct subheadings.

Reviewer 2

Title

Reviewer point #1: Consider shortening to “Genetic parameters of growth and efficiency traits in Menz sheep” for conciseness. I also saw an article on Dorper at the same research center by same author. “Efficiency-related traits” could be specified (e.g., Kleiber ratio, relative growth rate). Kleiber is not commonly used. Put research gap

Author response #1: We appreciate the reviewer’s suggestion. Our study specifically addresses growth rate and efficiency-related traits rather than growth traits. Although we previously investigated Dorper sheep, this work examines a different breed with distinct genetic characteristics and production systems. We clearly specify and describe the efficiency-related traits and the research gap in the background and subsequent sections of the manuscript. We therefore maintain that the current title is concise and accurately represents the study’s objectives.

Abstract

Reviewer point #2: L14–16: The phrase “traits related to growth efficiency are economically important and genetic improvement of these traits is vital” is slightly repetitive. Suggest: “Growth efficiency traits are economically important, and their genetic improvement is essential.”

Author response #2: Thank you for your correction. We have corrected in accordance with your comments

Reviewer point #3: L17–22: The abstract lists too many traits (6MW, ADG1, ADG2, ADG3, KR1, etc.) without context. This overwhelms the reader. Consider grouping (e.g., “average daily gain, Kleiber ratios, efficiency of growth, and relative growth rate”).

Author response #3: We respectfully disagree with the reviewer. The abstract must stand alone, and describing the studied traits is essential for clarity. We first grouped the average daily gain traits by growth period, while the other traits were presented accordingly to maintain a consistent and logical structure

Reviewer point #4: L29–32: “Direct additive genetic effect as the unique known random effect was the best fitting model …” → grammatically awkward. Suggest: “The model with direct additive genetic effect alone provided the best fit for most traits, except 6MW and ADG1, where maternal effects were also important.”

Author response #4: We sincerely thank the reviewer for the constructive comments. We have carefully considered the suggestions and made the corresponding corrections.

Reviewer point #5: L33–37: Heritability estimates are listed as a long series of numbers. Better to summarize ranges (e.g., “heritability ranged from 0.21 to 0.48 across traits”).

Author response #5: We thank the reviewer for the comments; however, we respectfully disagree. Presenting direct heritability estimates for all 13 studied traits as single value ranges would be misleading and inappropriate.

Reviewer point #6: L42–44: “Selection for 6MW would be more effective …” → Please clarify why six-month weight is the most informative trait. Already published so many times, repeated finding. Think new idea, a new suggestion that can advance the ongoing CBBP

Author response #6: We thank the reviewer for the constructive comments. The rationale for recommending selection based on 6MWT is already presented in the abstract. As 6MWT is positively correlated with the studied efficiency-related traits, we recommend using it as the primary selection criterion.

Reviewer point #7: Keywords: Good selection, but you may add “Menz sheep” explicitly as a keyword to improve discoverability.

Author response #7: Thank you for your suggestion. We have corrected in accordance with your comments

Introduction

Reviewer point #8: L49–56: Good background, but sentence “Small ruminants have been significantly supporting the livelihoods of smallholder farmers…” is awkward. Suggest: “Small ruminants play a major role in supporting the livelihoods of smallholder farmers across Ethiopia’s diverse ecological zones.

Author response #8: Thank you very much for your insightful comments and suggestions. We agree with the feedback and have revised the sentence accordingly.

Reviewer point #9: L57–66: The problem statement (low productivity despite adaptation) is strong. However, you need to clearly define the research gap: Few studies have estimated genetic parameters for efficiency traits (Kleiber ratio, RGR, GE) in Menz sheep.

Author response #9: We appreciate the reviewer’s valuable comments and have revised the sentence accordingly.

Reviewer point #10: L67–72: The introduction to Kleiber ratio is appropriate, but the rationale for including efficiency traits should be emphasized. Add supporting references from similar indigenous sheep studies.

Author response #10: Thank you for your feedback. We have made the necessary corrections based on your comments.

Reviewer point #11: Kleiber? I did not understand its importance

Author response #11: The Kleiber ratio (KR), calculated as growth rate relative to metabolic body weight, serves as an indirect measure of feed efficiency. In low-input, feed-scarce systems where Menz sheep are raised, recording individual feed intake is difficult. Using KR allows selection of animals that grow efficiently with limited feed, making it a practical and effective criterion for improving growth and feed efficiency while maintaining adaptation to harsh environments [15,17].

Reviewer point #12: L73–78: The objective statement is clear, but could be sharper: “This study aimed to estimate genetic parameters for growth rate and efficiency-related traits in Menz sheep under community-based breeding programs.”

Author response #12: We agreed with the reviewer and have made improvements as per the comments

Materials and methods

Reviewer point #13: Good detail about geography and climate. However, some redundancy (adaptability of Menz sheep already mentioned in Introduction). Suggest summarizing climatic info in fewer words, and moving references on adaptability to Introduction.

Author response #13: Thank you for your comments. We agree with your suggestion and have moved the statement on the adaptability of Menz sheep to the introduction section. However, since our focus is on efficiency traits, providing a detailed description of the breeding area, including its climatic conditions, remains important.

Reviewer point #14: L97: “New born lambs were weighted” → “weighed”.

Author response #14: Thank you for your correction. We have corrected accordingly.

Reviewer point #15: L97: L105: “…primarily focused on using six months weight as selection criteria” → “as the main selection criterion.” Since already described by other author just cite the previous authors. The procedure of CBBP selection is as described by

Author response #15: Thank you for your comment. We have cited the relevant authors; however, we retained a brief description of the selection criteria as it is important for clarity the ongoing breeding programs.

Reviewer point #16: L109–111: Provide clarity: “Rams were rotated annually between groups to minimize inbreeding.” Still duplication.

Author response #16: Thank you for your comment. It is not a duplication but rather an important detail specific to this section. We retained it to emphasize the role of annual ram rotation in minimizing inbreeding within the breeding program

Reviewer point #17: L117–118: Typo: ADG1 repeated twice (birth–weaning and weaning–6 months). One should be ADG2.

Author response #17: Thank you for your valuable comments. We have corrected.

Reviewer point #18: Equations should be formatted clearly with subscripts and consistent notation.

Author response #18: Thank you for your valuable comments. We have corrected accordingly.

Reviewer point #19: L133: Efficiency of growth formula is confusing; add units or explanation (percentage change relative to initial weight).

Author response #19: We agreed with the reviewer and provided the requested information as per the comments. However the clearly indicates the unit is percentage (%)

Reviewer point #20: L172: “…when the change in maximum log L between the last two iterations is less than 10-4” → please write as 1e-4 for clarity

Author response #20: Thank you for your comment. We would like to clarify that 10-4 follows standard mathematical notation and is widely preferred in scientific manuscripts. The notation '1e-4' is commonly used in programming and Excel contexts but is less formal in academic writing. Therefore, we have retained 10-4 in the text

Reviewer point #21: Genetic trend analysis (L190–192): Provide more detail on the time scale (2009–2023). Were all traits analyzed for trends or only key ones? Need clarification

Author response #21: Thank you for your comment. The genetic trends were estimated for the study period, as described in the Data Analysis section. Traits that did not exhibit improvement in genetic progress have not been presented in the figure as indicated in the Results section.

Results

Reviewer point #22: L193–207: Environmental effects are described well. However, Phrase “pedigree structures utilized in this study are fair enough to accurate genetic parameters” → awkward. Suggest: “The pedigree structure was adequate for accurate estimation of genetic parameters.”

Author response #22: We appreciate your comments and have revised the sentence.

Reviewer point #23: L208–217: Heritability values should again be summarized in ranges, with key examples. The full list can stay in the table.

Author response #23: We thank the reviewer for the suggestion. However, presenting heritability estimates in ranges across different traits is not meaningful, as the values differ substantially and such summarization could be misleading.

Reviewer point #24: L218–229: Correlations are strong, but the text is hard to follow. Suggest breaking into sub-sections: (i) within-period correlations, (ii) across-period correlations

Author response #24: We thank the reviewer for the suggestion. However, separating the correlations into sub-sections would not allow us to adequately present the correlations of traits across different growth periods, which is essential for understanding the overall relationships. Therefore, we have retained the original structure, describing both within- and across-period correlations together to provide a complete and interpretable overview.

Reviewer point #25: Figures 1–3: Ensure captions explain the units of genetic trend (kg/year, g/day/year, etc.).

Author response #25: We thank the reviewer for the comment. As shown in the figures, the annual genetic trend for 6MW is presented in kg, and for ADG in grams (g). For Kleiber ratio and growth efficiency, the units are percentages by definition, as indicated in the formulas, and therefore are not explicitly repeated in the figure captions. Since the figures show annual trends, it is implied that the units are per year.

Discussion

Reviewer point #26: L231–239: Good explanation of environmental effects. Could improve by citing more Ethiopian sheep literature beyond Baluchi/Kermani references

Author response #26: Thank you for your suggestion. We included all indigenous sheep breeds for which data on efficiency-related traits are available. However, studies on efficiency-related traits in Ethiopian sheep breeds remain limited

Reviewer point #27: L243–246: The hormonal explanation for sex differences is good, but speculative. Please cite a specific physiological reference

Author response #27: Thank you for your comments. We have already cited Ghafouri-Kesbi et al. (2017) and Mokhtari et al. (2019).

Reviewer point #28: L248–252: The finding that twins were more efficient pre-weaning but less efficient post-weaning is very interesting. This needs stronger discussion and reference support.

Author response #28: Thank you for your comments. As noted in the discussion section, our findings are contrary to those of similar studies. This discrepancy may be due to the relatively small number of twin-born lambs in our study. Which is included in the manuscript.

Reviewer point #29: L260–283: Discussion on heritability estimates is strong. However, some sentences are too long and repetitive. Simplify!

Author response #29: We agreed with reviewer and we improved the manuscript

Reviewer point #30: L285–307: The point about unfavorable pre- vs. post-weaning correlations is very important. Please emphasize its implication: selection should consider multi-trait indices to avoid negative correlated responses.

Author response #30: Thank you for your insightful comment. We acknowledge the importance of unfavorable pre- vs. post-weaning correlations in designing breeding strategies. While multi-trait selection indices can address such issues, their application is not always mandatory, particularly under smallholder breeding program conditions where implementation is often constrained. In our study, 6-month weight (6WT) showed positive correlations with most of the other growth traits. Thus, using 6WT as a single selection criterion would indirectly improve the other positively correlated traits. This makes single-trait selection based on 6WT a more practical and feasible approach for smallholder systems.

Conclusion

Reviewer point #31: L315–329: Well-structured, but slightly repetitive. Suggest trimming.

Author response #31: Thank you for your comments and we improved the conclusion.

Reviewer point #32: Add practical recommendation: e.g., “Breeding programs should prioritize six-month weight while monitoring efficiency-related traits to ensure balanced progress.” Too many times documented- see previous article including Bonga, Doyogena

Author response #32: We have already provided a practical recommendation that can be readily implemented in smallholder breeding programs to enhance overall breeding efficiency. Specifically, selection based on six-month weight (WT6) in Menz sheep is expected to improve the efficiency of the overall breeding program. To date, no studies have addressed efficiency-related traits in the Bonga and Doyogena sheep breeds.

Reference

Reviewer point #33: Generally good coverage, including Ethiopian studies. Some inconsistencies: L366: “and a pastoral system of ethiopia” → capitalize Ethiopia. Several references (e.g., #5) are poorly formatted and need editing. Please ensure consistent journal abbreviations (Small Ruminant Res. vs. Small Rumin Res.).

Author response #33: AThank you for your valuable comments. We have revised and improved the reference.

Tables & Figures

Reviewer point #34: Tables are informative but crowded. Suggest splitting if possible.

Author response #34: We appreciate the reviewer’s observation regarding the tables. While we acknowledge that the tables are detailed, we believe that presenting the results in the current format provides a more comprehensive and cohesive view, facilitating comparison across traits. Splitting the tables might reduce clarity and make interpretation more d

---

## [Editor Report · Decision Letter 1]

12 Nov 2025

Genetic evaluation of growth rate and efficiency-related traits in Menz sheep

PONE-D-25-45437R1

Dear Dr. Besufkad

We’re pleased to inform you that your manuscript has been judged scientifically suitable for publication and will be formally accepted for publication once it meets all outstanding technical requirements.

Kind regards,

Nesrein M. Hashem

Academic Editor

PLOS ONE

---

## [Editor Report · Acceptance letter]

PONE-D-25-45437R1

PLOS ONE

Dear Dr. Besufkad,

I'm pleased to inform you that your manuscript has been deemed suitable for publication in PLOS ONE. Congratulations! Your manuscript is now being handed over to our production team.

Kind regards,

on behalf of

Dr. Nesrein M. Hashem

Academic Editor

PLOS ONE